# A Prospective Observational Study on the Role of Immunohistochemical Expression of Orphanin in Laryngeal Squamous Cell Carcinoma Recurrence

**DOI:** 10.3390/jpm13081211

**Published:** 2023-07-30

**Authors:** Federico Sireci, Francesco Lorusso, Francesco Dispenza, Angelo Immordino, Salvatore Gallina, Pietro Salvago, Francesco Martines, Giuseppe Bonaventura, Maria Laura Uzzo, Giovanni Francesco Spatola

**Affiliations:** 1Otorhinolaryngology Section, Biomedicine, Neuroscience and Advanced Diagnosics Department, University of Palermo, Via del Vespro 129, 133, 90127 Palermo, Italy; federicosireci@hotmail.it (F.S.); dott.francescolorusso@gmail.com (F.L.); francesco.dispenza@gmail.com (F.D.); salvatore.gallina@unipa.it (S.G.); 2Audiology Section, Biomedicine, Neuroscience and Advanced Diagnosics Department, University of Palermo, Via del Vespro 129, 133, 90127 Palermo, Italy; pietrosalvago@libero.it (P.S.); francesco.martines@unipa.it (F.M.); 3Histology and Embriology Section, Biomedicine, Neuroscience and Advanced Diagnosics Department, University of Palermo, Via del Vespro 129, 133, 90127 Palermo, Italy; giuseppe.bonaventura@unipa.it (G.B.); marialaura.uzzo@unipa.it (M.L.U.); giovannifrancesco.spatola@unipa.it (G.F.S.)

**Keywords:** orphanin fq, neuropeptide, laryngeal cancer, locoregional neoplasm recurrence, squamous cell carcinoma

## Abstract

To date, histological biomarkers expressed by laryngeal cancer are poorly known. The identification of biomarkers associated with laryngeal squamous cell carcinoma (SCC), would help explain the tumorogenesis and prevent the possible recurrence of the lesion after treatment. For this reason, the aim of this study is to investigate, for the first time, the Orphanin expression in 48 human specimens of laryngeal SCC and evaluate its possible correlation with patients prognosis. We analyzed pathological specimens from 48 patients with laryngeal SCC to detect the presence of Orphanin by using an immunohistochemistry test. We compared the findings with healthy tissue acquired from patients who underwent surgery for mesenchymal benign tumours of the larynx. The specimens were stained with anti-Orphanin monoclonal antibodies. Results were processed through a computerised image analysis system to determine a scale of staining intensity. All the tumoural specimens examined showed a significant immunoreaction for Orphanin when compared with healthy tissues (*p* < 0.05) but with a different immune reactivity related to clinical-pathological features. A high Orphanin expression was not significantly related to Histological Grading (HG), TNM, and stage (*p* > 0.05). In the multivariate analysis, the Orphanin expression was significantly related only to the malignant recurrence (*p* < 0.05). Our study suggests that Orphanin could have a role in tumorigenesis by increasing the recurrence of cancer; therefore, it should be further explored as a possible biomarker for laryngeal cancer.

## 1. Introduction

Head and neck squamous cell carcinoma (HNSCC) accounts for 3% of all malignant tumours and affects more often men than women (i.e., sex ratio 2:1 to 4:1). Among HNSCC, squamous cell carcinoma (SCC) of the larynx represents the 20.8% and it is responsible for 85–95% of cancers affecting the larynx. The main risk factors for this histotype are tobacco and alcohol use, age, and viral infections (i.e., Papillomavirus).

Laryngeal squamous cell carcinomas (LSCCs) currently have a 5-year survival rate of 80–90% for stage I/II disease. The percentage of patients with LSCC alive, five years after the diagnosis decreases to 50% in the case of advanced LSCC, as these patients experience a recurrence following therapy [1,2].

Unfortunately, to date, molecular biomarkers expressed by laryngeal cancer are poorly known. The identification of biomarkers associated with laryngeal squamous cell carcinoma (SCC), would help explain the tumorigenesis, preventing the possible recurrence of the lesions and therefore improve tumour survival and disease-free rates after different therapeutic strategies (i.e., surgery, radiation therapy, and chemotherapy) [2].

In recent years, several neuropeptides have been implicated in tumours of other organs. Among these, the heptadecapeptide nociceptin/orphanin FQ (N/OFQ) seems to have a role in tumorigenesis but it has never been studied in laryngeal cancer.

FQ (N/OFQ) is a 17-aminoacid neuropeptide produced by peptidergic cells, that selectively interacts with opioid receptor-like 1 (ORL1), named N/OFQ peptide receptor (NOP) [3,4]. It is synthesised by specific neurons of the hypothalamus with monoaminergic function [5,6,7,8] but also by other cells in different healthy tissues such as the gastro-entero-pancreatic system (GEP), lung, and salivary glands [9,10,11]. For this reason, N/OFQ is involved in central and peripheral functions such as modulation of pain, stress, learning, memory, immunity, and orexic function. However, its presence in pathological conditions has not been fully defined [3,4].

The literature reports high plasmatic levels of N/OFQ in some liver conditions (i.e., Wilson disease and cirrhosis) and tumours (i.e., human neuroblastoma cell-lines and hepatocellular carcinoma), though its possible role in the pathogenesis of these conditions is not fully understood [12,13]. The most reliable data show that after binding its receptor, N/OFQ can activate lymphocytes by stimulating them to release cytokines involved in neo-angiogenesis; in addition, it seems that N/OFQ may activate complex intracellular mechanisms that promote tumour cells growth and inhibit their apoptosis [14,15].

The presence of N/OFQ in upper airway carcinoma, and specifically in larynx carcinoma, has not yet been thoroughly investigated; however, some studies on the role of some neuropeptides (e.g., leptin) in laryngeal cancer encouraged the research on this topic [16].

For these reasons, for the first time, we retrospectively evaluated the histopathological expression of N/OFQ in a series of laryngeal SCCs to better understand its possible role as a biomarker in the diagnosis and evaluation of a patient’s prognosis.

## 2. Materials and Methods

Sixty-four patients with laryngeal SCC were enrolled in this retrospective study. The study was approved by the Ethics Committee of our University Hospital “Paolo Giaccone” with the protocol number 11/2021. The main inclusion criterion was the eligibility for surgery as a first treatment, excluding inoperable patients and/or previously treated with neoadjuvant radiotherapy. Forty-eight patients (mean age of 62.3 + 13.53 years) were recruited (Table 1) and evaluated by fiberoptic laryngoscopy and a head and neck computerised tomography (CT) with and without contrast to study the size of the tumour (T) and the nodal involvement of the neck (N). They were then biopsied in suspension micro-laryngoscopy (DML). The biopsy specimens were sent to the Department of Human Pathology of our University Hospital which confirmed the diagnosis of laryngeal SCC in all the recruited cases. The patients underwent then to surgery and, in particular, twenty (41.6%) of them underwent a total laryngectomy, while Open Partial Horizontal Laryngectomy (OPHL) was performed in 28 (58.3%) patients, according to criteria of oncologic radicality in surgical therapy. In 44 (91.6%) cases, neck dissection was also performed. Twenty (41.6%) patients also underwent adjuvant chemo-radiotherapy. The healthy laryngeal epithelium was acquired from a control group of patients without HNSCC. These patients could be included in the control group as they underwent surgical removal of mesenchymal benign tumours of the larynx and these tissues were free from epithelial alterations typical of smokers (e.g., preneoplastic lesions). The follow-up of all patients was 36 months. Pathological staging, according to the Tumour, Lymph Node, Metastases (TNM) system of the Union for International Cancer Control (8th edition), and histological grade were determined according to the degree of differentiation of the tumour.

### 2.1. Histological Examination

The tissue was fixed in 10% formalin and was subjected to a wash cycle in H_2_O font after one day; they were then rinsed with distilled water, dehydrated with alcohol solutions of increasing grade (70%–95%–100%), and cleared in xylene before proceeding to the inclusion in paraffin. The sample was cut into sections and one of those was coloured in hematoxylin and eosin (HE), processed for immunohistochemical study by monoclonal antibodies for the N/OFQ and finally revealed by using detection kit En Vision + System-HRP with 3-Amino-9-Ethylcarbazole (AEC) as substrate (Dako).

Serial sections 5µ thick were cut with Leica microtome RM2145, dried overnight at 37 °C, and then stored at room temperature. The day after, slides were de-waxed and rehydrated by sequential immersion in a graded series of alcohols (100%–95%–70%) and transferred into water for 5 min; to inhibit any endogenous peroxidase activity, the slides were treated for five minutes with Peroxidase Block in hydrated incubation enclosure at room temperature. Subsequently, the sections were transferred in PBS (Na_2_HPO4, KH_2_PO_4_, KCl, NaCl pH 7.4–7.6). The following protocol was performed by using the En-Vision + System HRP kit with AEC as substrate (Dako). After rinsing with PBS for 4 min, the section was incubated overnight at 4 °C with polyclonal anti-N/OFQ (Ob A-20) (Chemicon International, Inc., Temecula, CA, USA) diluted 1:100. After the incubation, a wash with phosphate-buffered saline (PBS) was performed to remove any remaining antibodies. The section was incubated with Peroxidase labelled polymer conjugated to goat anti-rabbit immunoglobulin in Tris-HCl buffer which contained stabilizing protein and an antimicrobial agent. A new washing cycle (2× with PBS, 5 min each) was performed to remove the unbound polymer. Subsequently, AEC chromogen in substrate buffer was added for 5 min and then stopped in distilled water. The slide was removed from the water and one drop of aqueous mounting medium (DAKO Faramount) and a coverslip were applied. No counter-staining was carried out. The control was performed by omission of the primary antibody, and by incubating the section with antiserum saturated with homologous antigen.

The immunohistochemical specimen was studied with a Leica Laborlux S Microscope (Leica Microsystem GmbH Wetzlar, Germany) with a Nikon DSL2 photo digital system (Nikon Corp, Tokyo, Japan). As suggested by the literature in similar studies [16,17,18], we analyzed the sample, in double-blind. Moreover, the result was compared to an analysis system of images obtained from digital files. Images were elaborated by Adobe Photoshop CS6 (Adobe Systems Inc, San Jose, CA, USA). We used an objective lens with a magnification of 4xand we converted the colour profile from RGB to CMYK. The protein presence was detected by a yellow channel [19].

The images were processed by the software in a grey intensity scale according to colour luminance (i.e., from 0 to 256 grey values) [20].

The interpretation of immunoreactivity was based on the histochemical score (H-score) assessment by incorporating both the staining intensity (i) and the percentage of stained cells at each intensity level (Pi). The i values were rated on a scale from 0 to 3, and specifically: 0 (no evidence of staining), 1 (weak staining), 2 (moderate staining), and 3 (strong staining). The Pi values vary from 0% to 100%. The final H-score was derived from the sum of i multiplied by Pi. This score, therefore, was in a range from 0 to 300. An H-score greater than 200 was considered “high reactivity”, while an H-score of less than 200 was considered “low reactivity” [21].

### 2.2. Statistical Analysis

Statistical analyses were performed using the SPSS statistical software version 8.0 (SPSS Inc., Chicago, IL, USA). The Fisher’s exact test (FET) compared categorical variables and correlation with clinical data were evaluated by means of Spearman’s Rank Correlation Test (SRCT). The Kaplan-Meier estimator was used to evaluate the recurrences of the tumour. Multivariate logistic regression (MLR) analysis was used to establish the correlation between N/OFQ expression and pathological features (TNM, grading, and malignancy recurrence after treatment). A *p*-value of ≤0.05 was considered statistically significant.

## 3. Results

### 3.1. Clinical-Pathological and Histological Characteristics of the Patients

All the forty-eight cases in this study were males affected by SCC, with a mean age of 62.3 years (SD 13.5; range 43–85 years) (Table 1). OPHL was performed in 28 (58.3%) patients and total laryngectomy in 20 (41.6%) patients. Twelve, twenty-four, and twelve patients were respectively a T2, T3, and T4 pathological tumour stage. Selective neck dissection (SND) was performed in 44 (91.6%) patients and lymph node metastases were detected in 20 (41.7%) cases (pN+). Distant metastases were not found.

Eight (16.6%) patients had early cancer (Stage I or II) and 40 (83.4%) had advanced cancer (Stage III or IV). Eight, twenty, and twenty patients had G1, G2, and G3 grading, respectively.

Postoperative radiotherapy was indicated for 20 (41.6%) patients. During the follow-up time (36 months), 16 (33.3%) patients developed a recurrence with eight on T and eight on N, respectively, after a mean period of 14.25 months (SD 7.10 months).

Laryngeal pathological tissues showed a significant N/OFQ expression (*p* < 0.05) compared to healthy tissues in all cases (Table 2), specifically: 16 cases of low immune reactivity (33%), and 32 cases of high immune reactivity (66.6%). In particular, high expression was clear in the pathological epithelium (Figure 1). Cytoplasmatic peri nuclear staining was predominant in all tissues. Only 12 cases (25%) of the healthy control tissues showed a low immune reactivity. The expression of N/OFQ changed in relation to different clinic-pathological features.

### 3.2. N/OFQ and TNM Staging

Thirty-six (75%) patients with advanced stage (T3–T4) showed immune-reactivity for N/OFQ. Twenty-six (72.2%) showed a high expression N/OFQ without a significant difference compared to cases in the early stage (*p* = 0.20). All the 20 pN+ cases showed a high expression of N/OFQ. The SRCT did not show a correlation between the presence of N/OFQ in the T-isolated stage (rho = 0.36; *p* = 0.079) and the N-isolated stage (rho = 0.08; *p* = 0.36).

When we analysed the complete disease stage, 40 (83.3%) patients in an advanced stage (III–IV) showed immune-reactivity to N/OFQ. Thirty (75%) of these patients showed a high expression of N/OFQ but without any significant relation compared to the patient in the early stage (FET *p* = 0.36) (SRCT rho = 0.34; *p* = 0.097). MLR analysis showed that N/OFQ expression was not significantly related to TNM (*p* > 0.05).

### 3.3. N/OFQ and Grading

Forty (83.3%) patients with high grades (G2–G3) showed immune reactivity to N/OFQ. Thirty (75%) of these showed a high expression of N/OFQ but without any significant relation with patients with low grades (*p* = 0.68). SRCT did not show a correlation between N/OFQ expression and grading (rho = 0.34; *p* = 0.09). MLR analysis showed that N/OFQ expression was not significantly related to grading (odds ratio [OR] = 1.41; *p* = 0.17; 95% confidence interval [CI] −0.77–0.15).

### 3.4. N/OFQ and Recurrences

The sixteen cases with recurrence were associated with N/OFQ immune-reactivity (*p* = 0.66) but this expression, also compared with low levels, was not significant (Log-rank test, *p* > 0.05) (Figure 2). The SRCT did not show a correlation between N/OFQ and disease recurrence (rho = 0.27; *p* = 0.19). Multivariate logistic regression analysis showed that N/OFQ expression was significantly related to malignancy recurrence (odds ratio [OR] = 3.93; *p* = 0.0009; 95% confidence interval [CI] 1.68–4.75).

## 4. Discussion

The most frequently studied biomarkers of LSCC in the literature are long non-coding RNAs (lncRNAs), cell cycle regulators (Ki-67, cyclin D1, p27, p16, PCNA), apoptosis regulatory proteins (Bcl2), oncogenes and tumour suppressor genes (p53), molecules involved in growth factor pathways (EGFR, TGF-β), angiogenic (VEGF, angiogenin, CD105), structural (E-cadherin, CD44, osteopontin, FAK, and cortactin) and immunological (PD-L1) markers and sex hormones (ER, PR, AR, and PRLR).

lncRNAs in LSCC patients are associated with the pathological differentiation degree, worse clinical stages, cervical lymph node metastasis, reduced radiosensitivity, poor overall survival, and global worse prognosis.

With regards to cell cycle regulators, high levels of Ki-67 are associated with LCSS increased aggressiveness and worse prognosis; the rare laryngeal tumours P16-positive present instead a better response and improved survival after medical and ionising treatment, leading to an optimal apoptotic response to radiation and chemotherapy.

A high level of Bcl-2 represents a marker of radiotherapy’s failure because its overexpression is an index of reduced apoptosis.

Instead, p53 is an oncongene, frequent in HNSCC (60–80% of cases); unfortunately, only a few studies demonstrated a prognostic significance for p53 overexpression, with an association between this parameter, progression to malignancy, and poor patient outcome.

EGFR overexpression is an established negative prognostic factor also in LSCC: in fact, its expression is significantly higher in lesions that progress to malignancy, is correlated with the risk of metastasis to neck lymph nodes, and is overall associated with shorter survival [22]. On the other hand, the analysis of chemotherapy and radiotherapy success rates shows a statistically significant correlation with EGFR expression.

Among the angiogenetic markers, the most important marker is considered to be VEGF. Its expression increases with progression from mild to moderate to severe dysplasia, and carcinoma and it may be important as a predictor of adverse outcomes: in fact, also in LSCC, VEGF expression significantly correlates with local recurrence and/or metastases and shorter disease-free survival, especially in locoregionally advanced disease; moreover, it also seems to be a significant predictor of complete response to induction chemotherapy.

Regarding structural biomarkers, a lower level of E-cadherin expression seems to correlate with poor differentiation of the tumour, a major risk of nodal metastasis, advanced T, TNM stage, and shorter disease-free survival. In particular, in laryngeal and pharyngolaryngeal cancer, high CD44 expression is strongly associated with worse T, N, and differentiation grade, as well as poorer 5-year overall survival rate; in addition, CD44 is involved in the development and progression of laryngeal lesions and it may help to predict the risk of transformation of the benign or precancerous lesions to cancer.

The role of steroid hormones and steroid hormone receptors in laryngeal squamous cell carcinoma is not well understood and it remains controversial. Estrogens certainly are a potent mitogen stimulus, promoting invasion and metastasis in prostate, ovarian, breast, and uterine cells, but with regards to estrogen receptors as a potential prognostic factor in LSCC, the results are conflicting.

In recent years, it has been clarified that the immune system plays a central role in the control of tumour growth: indeed, it is responsible for an equilibrium state that blocks potentially invading cancer cells. In this light, it is then necessary to search for immunological biomarkers that can offer prognostic information and facilitate clinical decision-making, identifying subgroups of patients that are more likely to respond to immunotherapy.

In this sense, PD-L1, a transmembrane protein widely expressed on tumoural and immune cells as lymphocytes and macrophages in HNSCC, suppresses the adaptive arm of the immune system, downregulating T cell activation.

This rapid review of the several markers in laryngeal cancer showed that the different molecules may have a possible role but none of these is specific. Furthermore, the possible role of neuropeptides is poorly studied [17].

This study for the first time showed the immunohistochemical expression of N/OFQ in laryngeal SCC. According to the histochemical scoring (H-score) assessment, the lesions with scores less than 200 were considered low reactivity and the lesions with scores greater than 200 were considered high expression (Table 1).

In all tumour tissues examined, the expression of N/OFQ was demonstrated in 10 cases with low immune reactivity and 38 cases with high immune reactivity (*p* = 0.001) when compared to healthy tissues used as controls (Table 2).

On the one hand, the presence of 10 cases of low immunoreactivity reduces the statistical power of our tests, on the other hand, it is a clear demonstration of how the N/OFQ is highly expressed in those patients where laryngeal carcinoma exhibits a more aggressive and an increased likelihood of relapse on T or N.

We demonstrated the presence of this peptide in all the examined tumour specimens compared to the healthy control tissue samples (*p* = 0.001) but without any statistically significant difference between high and low expression, compared to the clinic-pathologic characteristics (*p* > 0.05).

No correlation was highlighted when considering the T, N, or complete TNM staging.

During the follow-up, 16 patients developed loco-regional recurrence; in all these cases we noticed a high expression of N/OFQ in tumour tissues without a significant difference between the patients with high levels of N/OFQ and those with low levels (Fisher Test, *p* = 0.66; Log-rank test, *p* > 0.05; Spearman’s rank correlation, rho = 0.27; *p* = 0.19).

However, multivariate logistic regression analysis showed that N/OFQ expression correlated with malignant recurrence (*p* < 0.05). We hypothesised that N/OFQ may be expressed in advanced cancer when a neoplastic transformation has already occurred since it plays a role in the survival and spreading of tumour cells. We also detected a high expression of N/OFQ in tumour cells infiltrating the connective tissue.

According to these data, we supposed that N/OFQ could be a possible histological marker to predict a possible recurrence of tumour.

As previously mentioned, at present, N/OFQ should be considered as a multifunctional neuropeptide expressed not just by organs involved in nutritional balance, but also in physiological and pathological tissues with a partially known function [5].

In physiological conditions, only Leone et al. hypnotised a possible role of leptin, orexin, and N/OFQ in the proliferation of sialocytes of salivary glands of some animals [23]. In pathological conditions, the presence of N/OFQ was evidenced as high in the tissue and plasma of hepatocellular carcinoma (HCC) where this neuropeptide activates NF-kB signalling to promote autophagy, which inhibited apoptosis in tumoural cells.

Unfortunately, high levels of N/OFQ are present in patients with hepatic diseases such as Wilson disease and liver cirrhosis. For this non-specific role, its application as a hepatic tumour marker cannot be reasonable [14,15].

Another malignant tumour in which N/OFQ plays a role is the glioblastoma. Bedini et al. performed a proteomic analysis that showed that the nociceptin receptor (NOPr) is significantly expressed in glioblastoma cells. Therefore, the authors evidenced a novel role of the peptide nociceptin as the endogenous ligand of the NOPr that regulates cell migration, proliferation, and increase of the glioblastoma cells [13]

Unfortunately, to the best of our knowledge, the presence of N/OFQ in the larynx has been scarcely investigated and for this reason, we cannot make a direct comparison with other authors. However, Gallina et al. studied the involvement of leptin in the laryngeal cancer [16,24]. This neuropeptide shares with N/OFQ the same origin as the neural crest cell and plays a role in the control of food intake. In their study, Gallina et al. highlighted a significant immune reactivity for leptin in all the specimens examined in the same ways as N/OFQ. In contrast to N/OFQ, leptin expression was significantly related to Grading only by SRCT and malignancy recurrence by SRCT, FET, and Kaplan-Meier product-limit estimate. The multivariate analysis showed that the recurrence was significantly related to N, HG, and leptin expression whereas N/OFQ showed only a significant correlation between its expression and malignant recurrence. According to the authors, leptin plays a role as a growth factor.

With regards to N/OFQ, the literature has not clarified its role in the cancer yet. According to some immunological studies, N/OFQ plays a new immunomodulatory role in the bidirectional control of the immune system: proinflammatory cytokines induce the expression of N/OFQ in nerve tissues which stimulates mast cell function and T lymphocytes to produce proinflammatory cytokines (i.e., IL-1, IL-6, TNF-alpha, TNF-gamma, prostaglandins, and endothelin) typically involved in angiogenesis and neoplastic growth. [25,26,27,28] Furthermore, in light of these results, leptin and N/OFQ, being both neuropeptides with an orexic and anorexic function, could explain the cachexia of cancer patients [29,30]. Another field of research in tumour genesis is the role of N/OFQ in the modulation of pain. N/OFQ is an opioid peptide and therefore the activation of its receptor induces the same intracellular events of other opioid receptors. Some studies demonstrated that cannabinoids exert an inhibitory effect on cancer growth by inducing apoptosis [31,32]. Caffarel et al. showed that Δ9–tetrahydrocannabinol stopped the G2–M transition of breast cancer cells [33]. Furthermore, tumour angiogenesis can be suppressed by cannabinoid treatment. In a recent study, Picardi et al. highlighted that anandamide, used in the treatment of breast cancer cells, inhibited endothelial proliferation by the downregulation of some factors, such as VEGF, leptin, and thrombopoietin [34]. The use of cannabinoids on the NOP receptor for the treatment of tumours has been poorly investigated. Only Sliepen et al. showed, an alleviation of cancer-induced bone pain in rats when using morphine hydrochloride trihydrate as the ligand of the NOP receptor [34,35]. Therefore, in light of the above-mentioned studies, as well as the evidence of N/OFQ expression in laryngeal cancer, we can only hypothesize a possible role of cannabinoids in the regulation of cancer pain and its onset in the larynx.

Our results, supported by the literature, show the role of N/OFQ in different tumours, and therefore suggest that this peptide could have a role in tumorigenesis by increasing the recurrence of cancer; its role as a possible biomarker for laryngeal cancer should be further explored.

## 5. Conclusions

Developing tools that support the early identification of patients with advanced SCC is crucial. For several tumours, molecular and histological biomarkers are used to differentiate those with advanced lesions who are more likely to experience recurrence. Our preliminary results indicate that N/OFQ is present in laryngeal SCCs and may be used as a marker to predict an increased risk of malignancy recurrence after therapy in patients. However, validation in a larger-scale series and the performance of PCR to demonstrate the neoplastic synthesis of this peptide is certainly needed.

## Figures and Tables

**Figure 1 jpm-13-01211-f001:**
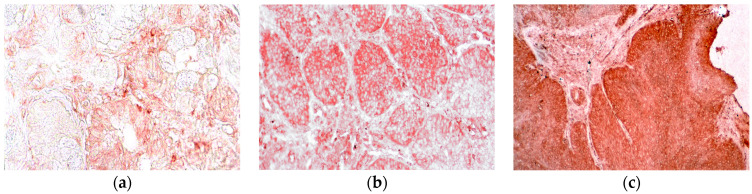
Absent (**a**), Low (**b**), and high (**c**) orphaninergic immunoreactivity in the epithelium of laryngeal cancer (20×).

**Figure 2 jpm-13-01211-f002:**
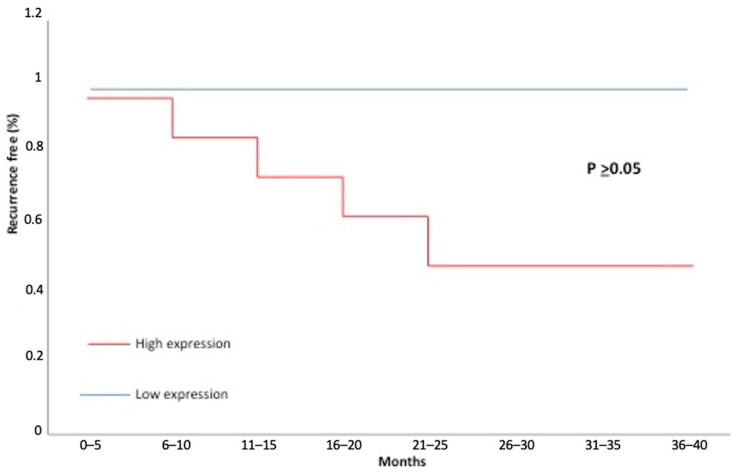
Kaplan-Meier recurrence-free curves categorised by high versus low expression of orphanin. The *p*-value was estimated by the log-rank test.

**Table 1 jpm-13-01211-t001:** Correlation between the clinicopathologic features of 48 patients with laryngeal squamous cancer and N/OFQ.

		N (%)	N/OFQ		*p* Value
			Low	High	
Cases		48	10 (20.8)	38 (79.2)	
Age	≤59	18 (37.5)	2 (11.1)	16 (88.8)	0.19
	>59	30 (62.5)	8 (26.6)	22 (73.3)	
T stadium (early/advanced)	T1–T2 (early)	12 (25)	0	12 (100)	0.20
	T3–T4 (advanced)	36 (75)	10 (27.7)	26 (72.2)	
pT1		0	0	0	
pT2		12 (25)	0	12 (100)	
pT3		24 (50)	8 (33.3)	16 (66.6)	
PT4		12 (25)	2 (16.6)	10 (83.3)	
N0 and N+ category	N0	28 (58.3)	8 (33.3)	20 (66.7)	0.11
	N+	20 (41.7)	2	18 (100)	
pN0		28 (58.3)	8 (28.6)	20 (71.4)	
pN1		4 (8.3)	2 (50)	2 (50)	
pN2		10 (25)	0	10 (100)	
pN3		4 (8.3)	0	4 (100)	
Tot			10 (20.8)	36 (79.2)	
Stage (early/advanced)	early (I–II)	8 (16.6)	0	8 (100)	0.36
	advanced (III–IV)	40 (83.3)	10 (25)	30 (75)	
Stage I		0	0	0	
Stage II		8 (16.6)	0	8 (100)	
Stage III		16 (33.3)	5 (31.2)	11 (68.8)	
Stage IV		24 (50)	5 (20.8)	19 (79.2)	
Grading (low and high)	Low (G1)	8 (16.6)	0	8 (100)	
	High (G2–G3)	40 (83.4)	10 (25)	30 (75)	
G1		8 (16.6)	0	8 (100)	0.36
G2		20 (41.6)	5 (25)	15 (75)	
G3		20 (41.6)	5 (25)	15 (75)	
Loco-regional recurrence	No	32 (66.6)	2 (6.2)	30 (93.8)	0.66
	Yes	16 (33.3)	0	16 (100)	

**Table 2 jpm-13-01211-t002:** N/OFQ immunoreactivity in cases and controls.

		Cases	Controls	*p* Value
N/OFQ	Yes	48	12	0.001
	No	0	36	

## Data Availability

Data supporting this study cannot be made available due to eyhical restrictions.

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
