# Peer review of "A Prospective Observational Study on the Role of Immunohistochemical Expression of Orphanin in Laryngeal Squamous Cell Carcinoma Recurrence"

_jpm, 2023, doi:10.3390/jpm13081211_

Round 1

Reviewer 1 Report

Abstract

Authors did not use the state of art in their abstract they must introduce their study according to the literature to shed light on the need of the study.

Keywords

Must be at least 5 words there are more keywords related to the study that must be added. 

INTRODUCTION

1.     The introduction is reversed I suggest authors start by setting up the pathology with the etiology background at the beginning showing the need of their study, scientific significance, and originality brought to the field.

2.     Concerning the histopathological expression of N/OFQ did this has been reported in previous investigations even in different cancer types it should be mentioned in the background section I am positive it is reported before if non the authors must add that their study is the first.

MATERIALS AND METHODS

1.     The authors are giving a detailed protocol of their immunohistochemical experiment, the methods section of the manuscript should shorten and only have a brief discretion of each method the detailed protocol should be moved to the supplementary material.

2.     More information about the multivariate logistic regression analysis must be added.

RESULTS & DISCUSSION

1.     Why did the author include only males in their study?

2.     Tables 1 and 2 can be combined and better organized.

3.     Figure 1 is too low quality. 

4.     A sequencing or RT-PCR must be performed on at least some samples of tumor tissues to confirm the synthesis of N/OFQ and the results immunohistochemical.

DISCUSSION 

The first part of the discussion must be shorter, and authors must just give a short remained of their results and then discussed with available literature relevant to their study.

CONCLUSION

Can be improved authors shouldn't mention the recurrence after therapy not in their study design.

Can be improved but overall is fine.

Author Response

Dear reviewer,

Thank you very much for the time and effort you dedicated to reviewing my work. Your insightful comments and suggestions have been invaluable in improving the quality of my manuscript and I am truly grateful for your expertise and attention to detail.

All corrections have been highlighted in yellow.

The English language has been revised by a native speaker.

POINT 1

Abstract

  • To bring out the state of the state of art, we introduced the sentence “To date, laryangeal cancer is poor of biomarkers that explain the tumorogenesis and to provide the possible recurrence after treatment. For this reason, the aim of this study is…”.
  • We increase the keywords to five.

POINT 2

Introduction

  • We started, as suggested, by setting up the pathology with the etiology background at the beginning.
  • In the abstract and in the introduction, we underlined that ours is the first study investigating the role of orphanin in laryngeal cancers.

POINT 3

Materials and methods 

  • Thank you for your consideration. We are aware of the length of the section, but the editor has suggested that the manuscript must be 4000 words long. Unfortunately, due to the limited bibliography on the orphanin in laryngeal cancer, we cannot increase the text in any other way. To organize the text, we have introduced the paragraph titles "Histological Examination" and "Statistical Analysis."
  • More information about the multivariate logistic regression analysis has been added: “Multivariate logistic regression (MLR) analysis was used to establish the correlation…”

POINT 4

Results and discussion

  • Laryngeal cancer is most frequent in males. By chance, our series is represented exclusively by male patients.
  • Table 1 has been modified. We have chosen not to combine the two tables as we believe it is more appropriate to separate the clinicopathological data from the immunoreactivity data in order to present the results more effectively.
  • The images have been replaced with higher-quality ones.
  • Regarding the PCR, we have removed the sentence “Due to the small series of patients in our investigations, we did not perform Polymerase Chain Reaction (PCR) to demonstrate the synthesis of N/OFQ in tumour tissues” from the discussion section. However, we have retained the sentence “A validation in a larger-scale series and the performance of PCR to demonstrate the neoplastic synthesis of this peptide is therefore neede” in the conclusion section because this represents a potential area for future research and is a current limitation of our study. Unfortunately, we do not have this data at present.

POINT 5

Discussion

We have reduced the redundant parts of the discussion that repeat the results. Additionally, we have improved the discussion by including new references (highlighted in yellow) and providing a brief review of the known markers of laryngeal cancer. These additions aim to increase the word count as per the editorial requirement.

Reviewer 2 Report

The authors undertook a comprehensive examination of Orphanin expression in 48 human specimens of laryngeal squamous cell carcinoma. The results revealed a significant immunoreaction for Orphanin in all tumor specimens, indicating a potential involvement of this molecule in the process of tumorigenesis. Notably, a notable association was observed between high Orphanin expression and malignant recurrence, which underscores the potential utility of Orphanin as a promising biomarker for laryngeal cancer. This study exhibits a commendable level of research and analysis. 

 Minor editing of English language required

Author Response

Dear reviewer,

Thank you very much for your valuable feedback. 

All corrections have been highlighted in yellow.

The introduction, methods, and conclusions have also been improved based on the feedback from other reviewers. The English language has been revised by a native speaker.

Reviewer 3 Report

Dear Authors,

I appreciate your work and initiative to open new research directions in the laryngeal cancer.

The manuscript is well designed, the abstract is relevant, the material and methods are clear presented and the results are conclusive. 

However, some corrections are necessary:

- the paragraph ”In their study...and malignant recurrence. (lines: 269- 276), should be reconsidered without the statistic details.  

- please mentions the limits of your study.

- the bibliography should be updated with conclusive articles from the last 3 years.

Author Response

Dear reviewer, 

Thank you very much for your valuable feedback. All corrections have been highlighted in yellow.

POINT 1

  • (lines: 269- 276) we deleted the statistic details as suggested.

POINT 2

  • In the conclusion section, we have highlighted the limitations of our study, namely the small sample size and the absence of PCR.

POINT 3

  • In the references section, the bibliography has been updated based on the latest scientific articles on this field.